# Multilabel Classification by Hierarchical Partitioning and Data-dependent Grouping

**Shashanka Ubaru**[*]
IBM Thomas J. Watson Research Center
Yorktown Heights, NY, USA

**Sanjeeb Dash**
IBM Thomas J. Watson Research Center
Yorktown Heights, NY, USA

**Arya Mazumdar**
Department of Computer Science
University of Massachusetts, Amherst, MA

**Oktay Gunluk**
Operations Research and Information Engg
Cornell Universty, Ithaca, NY

## Abstract

In modern multilabel classification problems, each data instance belongs to a small number of classes from a large set of classes. In other words, these problems involve learning very sparse binary label vectors. Moreover, in large-scale problems, the labels typically have certain (unknown) hierarchy. In this paper we exploit the sparsity of label vectors and the hierarchical structure to embed them in low-dimensional space using label groupings. Consequently, we solve the classification problem in a much lower dimensional space and then obtain labels in the original space using an appropriately defined lifting. Our method builds on the work of [35], where the idea of group testing was also explored for multilabel classification. We first present a novel data-dependent grouping approach, where we use a group construction based on a low-rank Nonnegative Matrix Factorization (NMF) of the label matrix of training instances. The construction also allows us, using recent results, to develop a fast prediction algorithm that has a *logarithmic runtime in the number of labels*. We then present a hierarchical partitioning approach that exploits the label hierarchy in large-scale problems to divide up the large label space and create smaller sub-problems, which can then be solved independently via the grouping approach. Numerical results on many benchmark datasets illustrate that, compared to other popular methods, our proposed methods achieve competitive accuracy with significantly lower computational costs.

## 1 Introduction

Multilabel classification (MLC) problems involve learning how to predict a (small) subset of classes a given data instance belongs to from a large set of classes. Given a set of labeled training data $\{x_i, y_i\}_{i=1}^n$ instances with input feature vectors $x_i \in \mathbb{R}^p$ and label vectors $y_i \in \{0,1\}^d$, we wish to learn the relationship between $x_i$s and $y_i$s in order to predict the label vector of a new data instance. MLC problems are encountered in many domains such as recommendation systems [16], bioinformatics [32], computer vision [9], natural language processing [26], and music [33]. In the large-scale MLC problems that we are interested in, the number of labels $d$ can be as large as $O(n)$ but the $\ell_0$-norm of the label vectors is quite small (constant). In some modern applications, the number of classes can be in the thousands, or even millions [40, 15]. However, the label vectors are typically sparse as individual instances belong to just a few classes. Examples of such large-scale MLC problems include image and video annotation for searches [39, 9], ads recommendation and web

---

[*]Corresponding Author. Email : `Shashanka.Ubaru@ibm.com`

page categorization [1, 29], tagging text and documents for categorization [34, 16], and others [15]. There are two practical challenges associated with these large-scale MLC problems: (1) how many classifiers does one have to train, and later, (2) what is the latency to predict the label vector of a new data instance using these classifiers. In the rest of this paper, we address these two challenges.

**Related Work:**   Most of the prior methods that have been proposed to solve large-scale sparse MLC problems fall under four categories:

(1) *One versus all (OvA) classifiers*: Earlier approaches for the MLC problem involve training a binary classifier for each label independently [46]. Recent approaches such as DiSMEC [2], PD-Sparse [43], PPD-Sparse [42], ProXML [3], and Slice [15] propose different paradigms to deal with the scalability issue of this naive approach. These methods typically train linear classifiers and achieve high prediction accuracy but at the same time suffer from high training and prediction runtimes. Slice reduces the training cost per label by subsampling the negative training points and reducing the number of training instances logarithmically.

(2) *Tree based classifiers*: These approaches exploit the hierarchical nature of labels when there is such a hierarchy, e.g., HOMER [34]. Recent tree based methods include FastXML [29], PfastreXML [16], Probabilistic Label Trees [18], Parabel [28], SwiftXML [27], extremeText [40], CraftXML [31], and Bonsai [20]. These methods yield high prediction accuracy when labels indeed have a hierarchical structure. However, they also tend to have high training times as they typically use clustering methods for label partitioning, and need to train many linear classifiers, one for each label in leaf nodes.

(3) *Deep learning based classifiers*: More recently, neural network based methods such as XML-CNN [24], DeepXML [47], AttentionXML [44], and X-BERT [7] have also been proposed. These methods perform as well as the tree based and OvA methods in many cases. However, they also suffer from high training and prediction costs, and the resulting model sizes can be quite large (in GBs).

(4) *Embedding based classifiers*: These approaches reduce the number of labels by projecting the label vectors onto a low-dimensional space. Most of these methods assume that the label matrix $Y$ is low-rank, see[32, 6, 48, 8, 45]. In this case, certain error guarantees can be established using the label correlation. However, the low-rank assumption does not always hold, see [5, 41, 2]. Recent embedding methods such as SLEEC [5], XMLDS [12] and DEFRAG [17] overcome this issue by using local embeddings and negative sampling. Most of these embedding methods require expensive techniques to recover the high-dimensional label vectors, involving eigen-decompositions or matrix inversions, and solving large optimization problems.

To deal with the scalability issue, a group testing based approach (MLGT) was recently proposed in [35]. This method involves creating $m$ random subsets (called groups defined by a binary group testing matrix) of classes and training $m$ independent binary classifiers to learn whether a given instance belongs to a group or not. When the label sparsity is $k$, this method requires only $m = O(k^2 \log d)$ groups to predict the $k$ labels and therefore, only a small number of classifiers need to be trained. Under certain assumptions, the labels of a new data instance can be predicted by simply predicting the groups it belongs to. The MLGT method has been shown to yield low Hamming loss errors. However, since the groups are formed in a random fashion, the individual classifiers might be poorly trained. That is, the random groupings might club together unrelated classes and the binary classifiers trained on such groups will be inefficient.

**Our contributions:**   In this work, we build on the MLGT framework and present a new MLC approach based on hierarchical partitioning and a data-dependent group construction. We first present the novel grouping approach (NMF-GT) that improves the accuracy of MLGT. This new method samples the group testing (GT) matrix (which defines the groups) from a low-rank Nonnegative Matrix Factorization (NMF) of the training data label matrix $Y = [y_1, \ldots, y_d]$. Specifically, we exploit symmetric NMF [22] of the correlation matrix $YY^T$, which is known to capture the clustering/grouping within the data [13]. This helps us capture the label correlations in the groups formed, yielding better trained classifiers. We analyze the proposed data-dependent construction and give theoretical results explaining why it performs well in MLGT. In the supplement, we discuss a GT construction that has constant weight across rows and columns, i.e., each group gets the same number of labels, and each label belongs to same number of groups. These constructions yield better classifiers and improved decoding, see Section 5 for details.

These new constructions also enable us – using recent results – to develop a *novel prediction algorithm* with *logarithmic* runtime in the number of labels $d$. If the sparsity of the label vector desired is $k$, then the complexity of the prediction algorithm will be $O(k \log \frac{d}{k})$. This significant improvement over existing methods will allow us to predict labels of new data instances in high-throughput and real-time settings such as recommendation systems [27]. This will address some of the limitations in traditional approaches to obtain related searches (search suggestions) [15].

We then present a hierarchical partitioning approach that exploits the label hierarchy in large-scale problems to divide the large label set into smaller subsets. The associated sub-problems can then be solved simultaneously (in parallel) using the MLGT approach. During prediction, the outputs of individual fast decoders are simply combined (or weighted) to obtain the top $k$ labels in log time. In numerical experiments, we first show that the new group construction (NMF-GT) performs better than the previous random constructions in [35]. We then compare the performance of the proposed hierarchical method (He-NMFGT) to some of the popular state-of-the-art methods on large datasets. We also show in the supplement, how the group testing framework can achieve *learning with less labeled data* for multilabel classification.

## 2 MLGT method

We first describe the group testing framework for MLC problems. The training data consists of $n$ instances $\{(x_i, y_i)\}_{i=1}^{n}$, where $x_i \in \mathbb{R}^p$ are the input feature vectors and $y_i \in \{0,1\}^d$ are the corresponding label vectors for each instance, and are assumed to be $k$-sparse, i.e., $||y_i||_0 \leq k$.

**Training.** The first step in training is to construct an $m \times d$ binary matrix $A$, called the *group testing* matrix. Rows of $A$ correspond to groups, columns to labels, and $A_{ij}$ is 1 if the $j$th label index (or class) belongs to the $i$th group. There exists an $A$ with $m = O(k \log d)$ (e.g., a $k$-disjunct matrix, see [35]) such that for any $k$-sparse binary vector $\tilde{y} \in \{0,1\}^d$, $\tilde{y}$ can be uniquely recovered (in polynomial time) from $\tilde{z} = A \vee \tilde{y}$. Here $\vee$ is the Boolean OR operation (replacing the vector inner product between a row of $A$ and $\tilde{y}$ in $A\tilde{y}$). In section 3, we describe how to construct these group testing matrices. This motivates projecting the label space into a lower-dimensional space via $A$, and creating *reduced* label vectors $z_i$ for each $y_i, i = 1, \ldots, n$ where $z_i = A \vee y_i$. The last step is to train $m$ binary classifiers $\{w_j\}_{j=1}^{m}$ on $\{x_i, (z_i)_j\}_{i=1}^{n}$ where $(z_i)_j$, the $j$th entry of $z_i$, indicates whether the $i$th instance belongs to the $j$th group or not. Algorithm 1 summarizes the training algorithm.

---

**Algorithm 1** MLGT: Training Algorithm

**Input:** Training data $\{(x_i, y_i)\}_{i=1}^{n}$, group testing matrix $A \in \mathbb{R}^{m \times d}$, binary classifier $\mathcal{C}$.
**Output:** $m$ classifiers $\{w_j\}_{j=1}^{m}$.
**for** $i = 1, \ldots, n$. **do**
$\quad z_i = A \vee y_i$.
**end for**
**for** $j = 1, \ldots, m$. **do**
$\quad w_j = \mathcal{C}(\{(x_i, (z_i)_j)\}_{i=1}^{n})$.
**end for**

**Algorithm 2** MLGT: Prediction Algorithm

**Input:** Test data $x \in \mathbb{R}^p$, the group testing matrix $A \in \mathbb{R}^{m \times d}$, $m$ classifiers $\{w_j\}_{j=1}^{m}$, sparsity $k$.
**Output:** predicted label $\hat{y}$.
**for** $j = 1, \ldots, m$. **do**
$\quad \hat{z}(j) = w_j(x)$.
**end for**
$\hat{y} = \text{fast-decode}(A, \hat{z}, k)$.

---

**Prediction.** For a new instance $x \in \mathbb{R}^p$, we first use the $m$ classifiers $\{w_j\}_{j=1}^{m}$ to predict a reduced label vector $\hat{z}$. We then apply the following simple linear decoding technique : For all $l \in [1, \ldots, d]$,

$$\hat{y}_l = \begin{cases} 1 & \text{if and only if } supp(A^{(l)}) \subseteq supp(\hat{z}) \\ 0 & \text{otherwise.} \end{cases}$$

Here, $supp(z) := \{i : z_i \neq 0\}$ denotes the support of the vector $z$. When $A$ is $k$-disjunct [35] and $\hat{z} = A \vee \hat{y}$ for some $k$-sparse vector $\hat{y}$, the above algorithm recovers $\hat{y}$. Unlike other embedding methods, this decoding technique does not require expensive matrix operations such as decompositions or inversion, and is linear in the number of labels $d$ using sparse matrix-vector products.

We will next present a new construction of $A$ together with a decoding algorithm that is logarithmic in $d$ and can be used in the last step of Algorithm 2 in place of the linear decoder described above.

# 3 Data dependent construction and decoding

In [35], the authors construct the group testing matrix $A$ using a uniform random construction that does not use any information about the training data. Even if two distinct classes (or label indices) are indistinguishable with respect to data instances, the columns of $A$ for these classes are different. We present a novel data-dependent construction for $A$ such that "similar" classes are represented by similar columns of $A$ and show that this construction leads to much better prediction quality. We also present a fast decoding technique. Consider the following metric:

$$\Phi_Y(A) = \|\frac{1}{n}YY^T - \frac{1}{m}A^TA\|_F, \tag{1}$$

$YY^T$ is the label correlation matrix, also called the label co-occurence matrix [20]. The $(i,j)$ entry of $YY^T$ is the number of training instances shared by the $i$th and $j$th classes. The entries of $A^TA$ give the number of groups shared by a pair of classes. Given a training label matrix $Y$, we construct $A$ so as to minimize $\Phi_Y(A)$, and have the groups membership structure for two similar classes be similar. See the supplement for relevant experiments. A completely random (disjunct) matrix is unlikely to yield low $\Phi_Y(A)$, since random grouping will not capture the correlation between labels. However, for proper decoding, the GT matrix needs to be sparse and columns need to have low coherence. We construct $A$ to account for both issues as follows.

Given $Y$ and $m$ – the number of groups – we compute a rank $m$ symmetric Nonnegative Matrix Factorization (symNMF) of $YY^T$ as $YY^T \approx H^TH$, where $H \in \mathbb{R}^{m \times d}$ is called the basis matrix [22]. It has been shown that symNMF is closely related to clustering, see [22, 13]. Given $Y$, the basis matrix $H$ defines the clustering within the labels. Therefore, we use the columns of $H$ to sample $A$.

For a column $h_i$ of $H$, let $\bar{h}_i$ be the normalized column such that its entries add to 1. Let $c$ be the column weights desired for $A$. For each column $i$, we form $\tilde{h}_i = c.\bar{h}_i$, and then re-weight these $\tilde{h}_i$ vectors in order to avoid entries $> 1$. We find all $\tilde{h}_i[j] > 1$, set these entries to 1 and distribute the excess sum $\sum(\tilde{h}_i[j] - 1)$ to the remaining entries. This is needed because many entries of $h_i$ will be zero. The columns of $A$ are then sampled using the re-weighted $\tilde{h}_i$s as the sampling probability vectors. Then each column will have $c$ ones per column on average. We do this instead of sampling the $i$th column of $A$ as a random binary vector – with the probability of the $j$th entry being 1 equal to $1/(k+1)$ – as in the $k$-disjunct construction used in [35] . In the supplement, we describe other constant weight constructions, where each group has the same number of labels, and each label belongs to same number of groups. Such constructions have been shown to perform well in the group testing problem [36, 38].

**Remark 1** (Choosing $c$)**.** *In these constructions, we choose the parameter $c$ (the column sparsity or the number of ones per column) parameter using a simple procedure. For a range of $c$s we form the matrix $A$, reduce and recover (a random subset of) training label vectors, and choose the $c$ which yields the smallest Hamming loss error.*

In MLGT, for our data-dependent GT matrix, we can use the linear decoder described in section 2. However, since the sampled matrix has constant weight columns, we can consider it as an adjacency matrix of a left regular graph. Therefore, we can use the recent proposed SAFFRON construction [23] and its fast decoding algorithm.

**Fast decoding algorithm via. SAFFRON:**  Recently, in [23], a biparitite graph based GT construction called SAFFRON (Sparse-grAph codes Framework For gROup testiNg) was proposed, see the supplement for details. Since our NMF based construction ensures constant weight columns, the resulting matrix $A$ can be viewed as an adjacency matrix of a left regular graph. This helps us adapt the fast decoding algorithm developed for the SAFFRON construction for label prediction in our method.

We next briefly describe the decoding algorithm (an adaptation of the fast decoder presented in [4] for sparse vector recovery). It has two steps, namely a *bin decoder* and a *peeling decoder*. The right nodes of the bipartite graph are called bins and the left nodes are called the variables.

Given the output reduced vector $z$ in the first step of prediction, the bin decoder is applied on to $m_1$ bins $A_i$'s (these are $m_1$ partitions of $A$ as per the construction, see supplement), and all the variable nodes connected to singletons (connected to non-zero nodes) are decoded and put to a set say $D$. Next,

in an iterative manner, a node from $D$ is considered at each iteration, and the bin decoder is applied to the bins connected to this variable node. If one of these node is a resolvable double-ton (connected to two nonzeros, but one already decoded), we can get a new nonzero variable ($y_i = 1$). These decoded variables are moved from $D$ to a new set of peeled off nodes $P$, and the newly decoded nonzero variable node, if any, is put in $D$. The decoder will terminate when $D$ is empty, and if the set $P$ has $k$ items, we have succeeded. For complete details, see [23]. The computational complexity of the decoding scheme is $O(k \log \frac{d}{k})$, see [38]. Therefore, for any left-regular graph with the SAFFRON construction and $m = O(k \log_2 d)$, the decoder recovers $k$ items in $O(k \log d)$ time. We can use this fast decoder in the last step of Algorithm 2 to predict the $k$ sparse label $\hat{y}$ for a given instance $x$.

**Analysis:** We next present an analysis that shows why the proposed data-dependent construction will perform well in MLGT. Let $\tilde{H}$ be the $m \times d$ reweighted matrix derived from the label data $Y$. $\tilde{H}$ is the potential matrix that is used to sample the $m \times d$ binary group testing matrix $A$. By construction, we know that the sum of entries in a column of $\tilde{H}$ is $c$, a constant.

Suppose in the prediction phase, the correct label vector is $y \in \{0, 1\}^d$. We know that there are at most $k$ ones in $y$, i.e., $|\operatorname{supp}(y)| \leq k$. Then, by using the $m$ binary classifiers we obtain the reduced label vector $z$, which if the classifiers are exact, will be $z = A \vee y$. To perform the decoding for $y$ then, in effect we compute $b = A^T z = A^T (A \vee y)$ and set the top $k$ coordinates to 1, the rest to 0. The next result shows the effectiveness of this method.

**Theorem 1** (Sampling $A$ using $Y$)**.** *For any $j \in \operatorname{supp}(y)$, $\mathbb{E}[b_j] = c$, whereas, for any $j \notin \operatorname{supp}(y)$, $\mathbb{E}[b_j] \leq \sum_{i=1}^{m} \exp(-\langle y, \tilde{h}^{(i)} \rangle)$, where $\tilde{h}^{(i)}$ is the ith row of $\tilde{H}$.*

The proof of this theorem is presented in the supplement. This result explains why our construction is a good idea. Indeed, since we generate $\tilde{H}$ in a data-dependent manner, any given label $y$ will likely have high correlations with the rows of $\tilde{H}$. As a result, the value of $b_j$ when $j$ is in the support of $y$ is much higher compared to the value of $b_j$ when $j$ is not in the support, with high probability. Therefore, choosing the top-$k$ coordinates of $b$ indeed will produce $y$.

## 4  Hierarchical approach for extreme classification

In very large-scale MLC problems (called extreme multilabel or XML problems), the labels typically have certain (unknown) hierarchy. By discovering and using this label hierarchy, one can design efficient classifiers for XML problems that have low computational cost. A limitation of our data-dependent approach is that we perform symNMF of the correlation matrix $YY^T$. As the symNMF problem is NP-hard, and also difficult to solve for matrices with more than a few thousand columns, getting good quality classifiers for XML problems is not guaranteed. Moreover, these large matrices are unlikely to be low rank [5]. Therefore, we propose a simple hierarchical label-partitioning approach to divide the set of label classes into smaller sets, and then apply our NMF-GT method to each smaller set independently.

Matrix reordering techniques on sparse matrices are popularly used for graph partitioning [19] and solving sparse linear systems [30]. Here, a large sparse matrix (usually the adjacency matrix of a large graph) is reordered such that the matrix/graph can be partitioned into smaller submatrices that can be handled independently. Since the label matrix $Y$ is highly sparse in XML problems and the labels have a hierarchy, the nonzero entries in $YY^T$ can be viewed as defining an adjacency matrix of a sparse graph. Let $G = (V, E)$ denote a graph, where each node corresponds to a label, and $e = ij \in E$ if and only if $YY_{ij}^T \neq 0$. In other words, an edge between nodes/labels $i$ and $j$ is present if and only if labels $i$ and $j$ occur together in at least one data point, which indicates "interaction" between these labels.

Suppose that $G$ has say $\ell$ components, i.e., it can be partitioned into $\ell$ disjoint sub-graphs, as assumed in Bonsai [20]. Then each component corresponds to a subset of labels that interact with one another but not with labels in other components. Permuting the labels so that labels in a component are adjacent to one another, and applying the same permutation to the columns of $Y$, one can obtain a block-diagonal reordering of the label matrix $YY^T$. Now the symNMF problem for $YY^T$ can be reduced to a number of smaller symNMF problems, one for each block of the matrix. Most large datasets (label matrices) with hierarchy will have many smaller non-interacting subsets of labels and

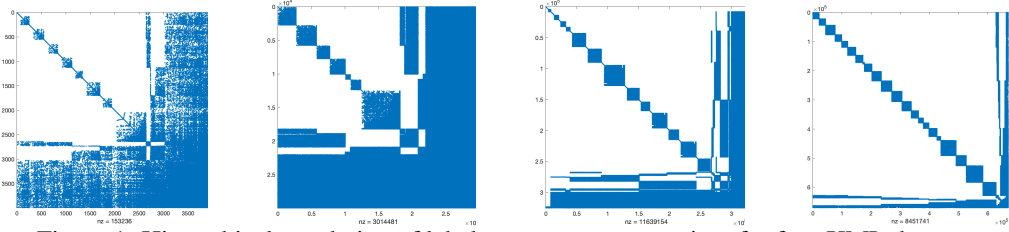
Figure 1: Hierarchical reordering of label co-occurrence matrices for four XML datasets.

few subsets that interact with many other labels. A natural approach is to use the vertex separator partitioning based reordering [11] or nested dissection [19] to obtain this permutation.

The idea is to find a small vertex separator $S$ of $G$ (here $S \subset V$) such that $G \setminus S$ has a number of disjoint components $C_1, \ldots, C_\ell$. The labels can then be viewed as belonging to one of the subsets $S \cup C_1, \ldots, S \cup C_\ell$, and we can apply NMF-GT to each separately. This idea can be further extended to a hierarchical partitioning of $G$ (by finding partitions of the subgraphs $C_i$ as $S_i, C_{i1}, \ldots, C_{i\ell}$ − where $S_i$ is a vertex separator of $C_i$). Each level of the hierarchy would be partitioned further till the components are small enough so that the MLGT (sym-NMF) algorithm can be efficiently applied.

In Figure 1, we display the hierarchical reordering of $YY^T$ obtained by the algorithm in [11] for four popular XML datasets: Eurlex (with number of labels $d \approx 4K$), Wiki10 ($d \approx 30K$), WikiLSHTC ($d \approx 325K$), and Amazon ($d \approx 670K$), respectively. We note that there are a few distinct blocks $C_i$ (the block diagonals), where the labels only occur together and are independent of other blocks (do not interact). We also have a small subset of labels $S$ (the outer band) that interact with most blocks $C_i$. We can partition the label set into $\ell$ subsets $\{S \cup C_i\}_{i=1}^\ell$ of size $\{d_i\}_{i=1}^\ell$ each and apply our NMF based MLGT individually (it can be done in parallel). During prediction, the individual fast decoders will return the positive labels for each subset in $O(\log d_i)$ time. We can simply combine these positive labels or weight them to output top $k$ labels. Since the subset $S$ of labels interact with most other labels and occur more frequently (power-law distribution), we can rank them higher when picking top $k$ of the outputted positive labels.

**Comparison with tree methods:** The tree based methods such as HOMER [34], Parabel [28], Bonsai [20], and others use label partitioning to recursively construct label tree/s with pre-specified number of labels in leaf nodes or at a tree depth. Most methods use k-means clustering for partitioning (typically clustering is used to partition at each non-leaf node), and this can be expensive for large $d$. Then OvA classifiers are learned for each label in the leaf nodes. However, in our approach, we use label partitioning to identify label subsets on which we can apply NMF-GT independently. Our matrix reordering approach is inexpensive with cost $O(nnz(YY^T)) = O(dk)$, see [11]. We use the NMF-GT strategy to learn only $O(k \log d_i)$ classifiers per partition.

## 5 Numerical Experiments

We now present numerical results to illustrate the performance of the proposed approaches (the data-dependent construction NMF-GT and with hierarchical partitioning He-NMFGT) on MLC problems. Several additional results and details are presented in the supplement.

**Datasets:** For our experiments, we consider some of the popular publicly available multilabel datasets put together in *The Extreme Classification Repository* [5] (http://manikvarma.org/downloads/XC/XMLRepository.html). The applications, details and the original sources of the datasets can be found in the repository. Table 1 lists the statistics.

Table 1: Dataset statistics

| Dataset | $d$ | $k$ | $n$ | $nt$ | $p$ |
|---|---|---|---|---|---|
| Mediamill | 101 | 4.38 | 30993 | 12914 | 120 |
| Bibtex | 159 | 2.40 | 4880 | 2515 | 1839 |
| RCV1-2K (ss) | 2016 | 4.76 | 30000 | 10000 | 29699 |
| EurLex-4K | 3993 | 5.31 | 15539 | 3809 | 5000 |
| AmazonCat(ss) | 7065 | 5.08 | 100000 | 50000 | 57645 |
| Wiki10-31K | 30938 | 18.64 | 14146 | 6616 | 101850 |
| WikiLSHTC | 325056 | 3.18 | 1813391 | 78743 | 85600 |
| Amazon-670K | 670091 | 5.45 | 490449 | 153025 | 135909 |

In the table, $d = $ #labels, $\bar{k} = $ average sparsity per instance, $n = $ # training instances, $nt = $ # test instances and $p = $ #features. The datasets marked (ss) are subsampled versions of the original data with statistics as indicated.

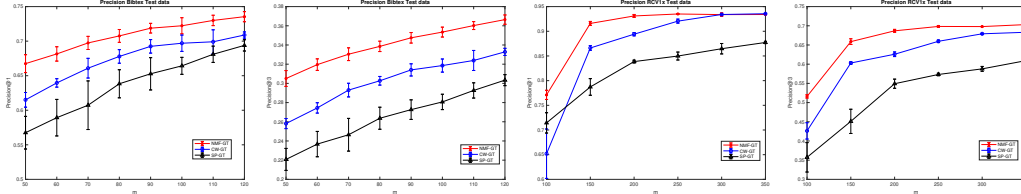

Figure 2: $\Pi@1$ and $\Pi@3$ for test data instances for bibtex (left two) and RCV1x (right two) datasets as a function of number of groups $m$. Error bar over 10 trials.

**Evaluation metrics:** For comparison, we use the popular evaluation metric called $Precison@k$ (P@k) [1] and a modification $\Pi@k$ with $k = \{1, 3, 5\}$, defined as

$$P@k := \frac{1}{k}\sum_{l \in rank_k(\hat{y})} y_l \quad \text{and} \quad \Pi@k := \frac{1}{k}\min\left(k, \sum_{l \in top_5(\hat{y})} y_l\right), \tag{2}$$

respectively. We use $\Pi@k$ since MLGT returns binary vectors, see details in the supplement.

**Comparing group testing constructions:** In the first set of experiments, we compare the new group testing constructions with the sparse random construction (SP-GT) used in [35], where each entry of $A$ is sampled with uniform probability $prob = \frac{1}{k+1}$. Our first construction (NMF-GT) is based on the symNMF as described in Section 3. Given the training label matrix $Y$, we first compute the symNMF of $YY^T$ of rank $m$ using the Coordinate Descent algorithm by [37] (code provided by the authors) and then compute a sparse binary matrix using reweighted rows of the NMF basis. Our second construction (CW-GT) is the constant weight construction defined in supplementary A.1. For both constructions, the number of nonzeros (sparsity) per column of $A$ is selected using the search method described in Remark 1, see supplement for more details.

Figure 2 plots $\Pi@1$ and $\Pi@3$ we obtained for the three constructions (we depict NMF-GT, CW-GT and SP-GT by red stars, blue circles, and black triangles, respectively) as the number of groups $m$ increases. See the supplement for an enlarged version. The first two plots correspond to the Bibtex dataset, and the next two correspond to RCV1x. As expected, the performance of each construction improves as the number of groups increases. Note that NMF-GT consistently outperforms the other two. In the supplement, we compare the three constructions (for accuracy and runtime) on four datasets. We also include the One versus All (OvA) method (which is computationally expensive) to provide a frame of reference.

In all cases, NMF-GT outperforms the other two (possibly because it groups the labels based on the correlation between them), and CW-GT performs better than SP-GT. Both NMF-GT and CW-GT ensure that $m$ classifiers are trained on similar amounts of data. Decoding will also be efficient since all columns of $A$ have the same support size. NMF-GT is superior to the other two constructions, and therefore, we will use it in the following experiments for comparison with other popular XML methods.

**Comparison with popular methods:** We next compare the NMF-GT method (best one from our previous experiments) and the hierarchical method (He-NMFGT) with four popular methods, namely MLCS [14], SLEEC [5], PfastreXML [16], and Parabel [28] with respect to the modified precision $\Pi@k$ metric. Table 2 summarizes the results obtained by these six methods for different datasets along with total computation time $T_{total}$ and the test prediction time $T_{test}$. The no. of groups $m$ used in NMFGT and no. of blocks $\ell$ used in He-NMFGT are also given.

We note that NMF-GT performs fairly well given its low computational burden. The hierarchical approach He-NMFGT yields superior accuracies with similar runtimes as NMFGT (outperforms other methods wrt. $\Pi@1$). PfastreXML and Parabel yield slightly more accurate results in some cases, but require significantly longer run times. Note that the prediction time $T_{test}$ for our methods are orders of magnitude lower in some cases. For He-NMFGT, $T_{total}$ includes computing the partition, applying MLGT for one block (since this can be done in parallel), and predicting the labels of all test instances. For smaller two datasets, He-NMFGT was not used since they lacked well-defined partitions. In He-NMFGT, for the labels shared across partitions, we use weights for the label outputs such that these weights add to 1. The matrix reordering method recursively partitions the labels, hence discovering a hierarchy. The code we use produces the partitions (and sizes), in addition to the permutations depicted in Fig. 1.

Table 2: Comparisons between different MLC methods. Metric: Modified Precision

| Dataset | Metrics | He-NMFGT | NMF-GT | MLCS | SLEEC* | PfastreXML | Parabel |
|---|---|---|---|---|---|---|---|
| Mediamill | $\Pi$@1 | – | 0.8804 | 0.8359 | 0.8538 | **0.9376** | 0.9358 |
| $d = 101$ | $\Pi$@3 | – | 0.6069 | 0.6593 | 0.6967 | **0.7701** | 0.7622 |
| $m = 50$ | $\Pi$@5 | – | 0.3693 | 0.4102 | **0.5562** | 0.5328 | 0.5169 |
| | $T_{total}$ | – | **17.2s** | 20.3s | 3.5m | 190.1s | 74.19s |
| | $T_{test}$ | – | **0.17s** | 6.93s | 80.5s | 18.4s | 17.85s |
| RCV1x | $\Pi$@1 | – | 0.9350 | 0.9244 | 0.9034 | 0.9508 | **0.9680** |
| $d = 2016$ | $\Pi$@3 | – | 0.6983 | 0.6945 | 0.6395 | 0.7412 | **0.7510** |
| $m = 250$ | $\Pi$@5 | – | 0.4502 | 0.4486 | 0.4457 | 0.4993 | **0.5040** |
| | $T_{total}$ | – | **88.4s** | 541.1s | 34m | 7.73m | 6.7m |
| | $T_{test}$ | – | **1.04s** | 176.7s | 53.1s | 3.03m | 1.68m |
| Eurlex | $\Pi$@1 | **0.9265** | 0.8477 | 0.8034 | 0.7474 | 0.9004 | 0.9161 |
| $d = 3993$ | $\Pi$@3 | 0.7084 | 0.5547 | 0.5822 | 0.5885 | 0.6946 | **0.7397** |
| $m = 350$ | $\Pi$@5 | 0.4807 | 0.3444 | 0.3965 | 0.4776 | 0.4939 | **0.5048** |
| $\ell = 4$ | $T_{total}$ | 322s | **227.3s** | 343.3s | 21m | 11.8m | 6.1m |
| | $T_{test}$ | 1.1s | **0.94s** | 235.1s | 45s | 59.2s | 74.3s |
| Amazon13 | $\Pi$@1 | **0.9478** | 0.8629 | 0.7837 | 0.8053 | 0.9098 | 0.9221 |
| $d = 7065$ | $\Pi$@3 | 0.6555 | 0.5922 | 0.5469 | 0.5622 | 0.6722 | **0.6957** |
| $m = 700$ | $\Pi$@5 | 0.4474 | 0.3915 | 0.3257 | 0.4152 | 0.5119 | **0.5226** |
| $\ell = 4$ | $T_{total}$ | 8.7m | **7.5m** | 19.7m | 68.8m | 27.5m | 16.9m |
| | $T_{test}$ | 4.42s | **4.21s** | 13.7m | 106.3s | 241.6s | 114.7s |
| Wiki10 | $\Pi$@1 | **0.9666** | 0.9155 | 0.5223 | 0.8079 | 0.9289 | 0.9410 |
| $d = 30938$ | $\Pi$@3 | **0.7987** | 0.6353 | 0.2995 | 0.5050 | 0.7269 | 0.7880 |
| $m = 800$ | $\Pi$@5 | **0.5614** | 0.4105 | 0.1724 | 0.3526 | 0.5061 | 0.5502 |
| $\ell = 5$ | $T_{total}$ | 14.7m | **13.6m** | 63m | 54.9m | 40.5m | 33.5m |
| | $T_{test}$ | 11.5s | **9.82s** | 45m | 51.3s | 8.2m | 4.2m |

Table 3: Comparisons between different XML methods. Metric: Standard Precision

| Dataset | Metrics | Embedding | | Tree | | | OvA | | DNN |
|---|---|---|---|---|---|---|---|---|---|
| | | He-NMFGT | SLEEC | PfastreXML | Parabel | XT | Dismec | PPD-sparse | XML-CNN |
| Eurlex | $P$@1 (%) | 75.04 | 74.74 | 73.63 | 74.54 | – | 83.67 | 83.83 | 76.38 |
| $d = 3993$ | $P$@3 (%) | 61.08 | 58.88 | 60.31 | 61.72 | – | 70.70 | 70.72 | 62.81 |
| $\ell = 4$ | $P$@5 (%) | 48.07 | 47.76 | 49.39 | 50.48 | – | 59.14 | 59.21 | 51.41 |
| $m = 300$ | $T_{train}$ | **4.8m** | 20m | 10.8m | 5.4m | – | 0.94hr | 0.15hr | 0.28hr |
| | $T_{test}/nt$ | **0.28ms** | 4.87ms | 1.82ms | 0.91ms | – | 7.05ms | 1.14ms | 0.38ms |
| Wiki10 | $P$@1 (%) | 82.28 | 80.78 | 82.03 | 83.77 | 85.23 | 85.20 | 73.80 | 82.78 |
| $d = 30938$ | $P$@3 (%) | 69.68 | 50.50 | 67.43 | 71.96 | 73.18 | 74.60 | 60.90 | 66.34 |
| $\ell = 5$ | $P$@5 (%) | 56.14 | 35.36 | 52.61 | 55.02 | 63.39 | 65.90 | 50.40 | 56.23 |
| $m = 650$ | $T_{train}$ | **14.2m** | 53m | 32.3m | 29.3m | 18m | – | – | 88m |
| | $T_{test}/nt$ | **0.69ms** | 7.7ms | 74.1ms | 38.1ms | 1.83ms | – | – | 1.39s |
| WikiLSHTC | $P$@1 (%) | 55.62 | 54.83 | 56.05 | 64.38 | 58.73 | 64.94 | 64.08 | – |
| $d = 325056$ | $P$@3 (%) | 33.81 | 33.42 | 36.79 | 42.40 | 39.24 | 42.71 | 41.26 | – |
| $\ell = 12$ | $P$@5 (%) | 23.04 | 23.85 | 27.09 | 31.14 | 29.26 | 31.5 | 30.12 | – |
| $m = 800$ | $T_{train}$ | **47.5m** | 18.3hr | 7.4hr | 3.62hr | 9.2hr | 750hr | 3.9hr | – |
| | $T_{test}/nt$ | **0.8ms** | 5.7ms | 2.2ms | 1.2ms | **0.8ms** | 43m | 37ms | – |
| Amazon670K | $P$@1 (%) | 39.60 | 35.05 | 39.46 | 43.90 | 39.90 | 45.37 | 45.32 | 35.39 |
| $d = 670091$ | $P$@3 (%) | 36.78 | 31.25 | 35.81 | 39.42 | 35.60 | 40.40 | 40.37 | 33.74 |
| $\ell = 22$ | $P$@5 (%) | 32.40 | 28.56 | 33.05 | 36.09 | 32.04 | 36.96 | 36.92 | 32.64 |
| $m = 800$ | $T_{train}$ | **47.8m** | 11.3hr | 1.23hr | 1.54hr | 4.0hr | 373hr | 1.71hr | 52.2hr |
| | $T_{test}/nt$ | **1.45ms** | 18.5ms | 19.3ms | 2.8ms | 1.7ms | 429ms | 429ms | 16.2ms |

In Table 3 we compare the performance of He-NMFGT with several other popular XML methods wrt. the standard $P$@$k$ metric. We compare the accuracies and computational costs for He-NMFGT, SLEEC (embedding method), three tree methods (PfastreXML, Parabel, ExtremeText XT), two OvA methods (Dismec, PD-sparse) and a DNN method XML-CNN (see sec. 1 for references). The precision results and the runtimes for the four additional methods were obtained from [28, 40]. In the table a '–' indicate these results were not reported by the authors.

We note that, compared to other methods, He-NMFGT is significantly faster in both training and test times, and yet yields comparable results. The other methods have several parameters that need to be tuned. More importantly, the main routines of most other methods are written in C/C++, while He-NMFGT was implemented in Matlab and hence we believe the run times can be improved to enable truly real-time predictions. The code for our method is publicly available at https://github.com/Shashankaubaru/He-NMFGT. Several additional results, implementation details and result discussions are given in the supplement.

**Trade off and improvements:** In the above two tables, the precision and runtime results we present are based on an optimal trade-off (between accuracy and runtimes) we obtained by our method. In particular, the results reported in the tables are for the smallest $m$ for which our accuracy is close to

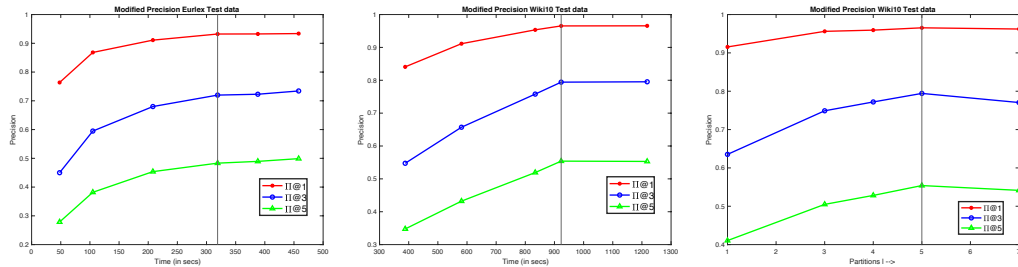

Figure 3: Trade off: Precision ($\Pi$@1, $\Pi$@3, $\Pi$@5) as functions of runtimes (in secs) for Eurlex (left) and wiki10 (middle) dataset, and no. of partitions $\ell$ (right).

the SOA tree methods. In figure 3, we plot the precision values ($\Pi$@1, $\Pi$@3, $\Pi$@5) as a function of the runtimes (in secs) for Eurlex (left) and wiki10 (middle) datasets, by increasing # groups $m$ in each partition. Indeed, we notice a clear trade-off: as we increase runtimes, accuracy improves. But beyond a point, the accuracy gain is limited as $m$ is increased. Improved accuracy can be achieved for higher runtimes (when $m$ is much more than $k \log d$). We also plot $\Pi$@$k$ versus # partitions $\ell$ for Wiki10 (right). For smaller $\ell$, it is hard to compute a good NMF for large matrices, and with many partitions, we will miss certain label correlations. In the tables above, we report the precision numbers – marked by the black vertical lines in the plots – that yield the best trade-offs.

## Conclusions

In this paper, we extended the MLGT framework [35] and presented new GT constructions (constant weight and data dependent), and a fast prediction algorithm that requires logarithmic time in the number of labels $d$. We then presented a hierarchical partitioning approach based on matrix reordering techniques to scale the MLGT approach to larger datasets. Our computational results show that the NMF construction yields superior performance compared to other GT matrices. We also presented a theoretical analysis which showed why the proposed data dependent method (with a non-trivial data-dependent sampling approach) will perform well. With a comprehensive set of experiments, we showed that our method is significantly faster in both training and test times, and yet yields competitive results compared to other popular XML methods.

## Broader Impact

Our work presents a novel approach to solve the extreme multilabel (XML) classification problem. The main contributions of our paper are (a) the use of matrix reordering techniques to hierarchically partition the label space, and (b) the development of a data-dependent group testing scheme, that improves label grouping significantly for MLGT, and can leverage the recently proposed log-time decoding algorithm. These innovations lead to a significantly faster training algorithm compared to most existing methods that yields comparable results. The XML problem is encountered in a number of applications from related searches to ad recommendations to natural language understanding in the technology industry, and from gene and molecule classifications to learning neural activities in scientific data analysis. A fast algorithm like ours will enable prediction in high-throughput and real-time settings, and address some of the limitations in traditional inline search suggestions or approaches for related searches. The preliminary results presented in the supplement demonstrate how the group testing approach achieves *learning with less data* for MLC. There is increasing interest among research (and defense) agencies in developing learning algorithms that achieve 'Learning with Less Labels' (LwLL), see `darpa.mil/program/learning-with-less-labels`. Thus, the business and research impacts are likely to be significant (and clear).

**Acknowledgment:** We thank the anonymous reviewers for their valuable comments and suggestions.

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
