[Supplementary Material]

# Supplementary material - Multilabel Classification by Hierarchical Partitioning and Data-dependent Grouping

## A  Constant Weight Construction

In this supplement, we first describe two constant weight constructions, where each group has the same number of labels, and each label belongs to the same number of groups. Such constructions have been shown to perform well in the group testing problem [36, 38].

### A.1  Randomized construction

The first construction we consider is based on LDPC (low density parity) codes. Gallagher proposed a low density code with constant weights in [10]. We can develop a constant weight GT matrix $A$ based on this LPDC construction as follows: Suppose the matrix $A$ we desire has $d$ columns with constant $c$ ones in each column, and $r$ ones in each row. The LDPC matrix will have $dc/r$ rows in total. The matrix is divided into $c$ submatrices, each containing a single 1 in each column. The first of these submatrices contains all the ones in descending order, i.e., the $i$th row will have ones in the columns $(i-1)r+1$ to $ir$. The remaining submatrices are simply column permutations of the first. We consider this construction in our experiments.

### A.2  SAFFRON construction

Recently, in [23], a biparitite graph based GT construction called SAFFRON (Sparse-grAph codes Framework For gROup testiNg) was proposed. [38] extended this SAFFRON construction to form left-and-right-regular sparse-graph codes called regular-SAFFRON. The adjacency matrices corresponding to such graphs give us the desired constant weight constructions. The regular-SAFFRON construction starts with a left-and-right-regular graph $G_{cr}(d, m_1)$, with $d$ left nodes called variable nodes, and $m_1$ right nodes called bin nodes. The $d.c$ edge connections from the left and $m_1.r$ edge connections from the right are paired up according to a random permutation.

Let $T_G \in \{0, 1\}^{m_1 \times d}$ be the adjacency matrix corresponding to the left-and-right-regular graph $G_{cr}(d, m_1)$. Then, $T_G$ has $c$ ones in each column and $r$ ones in each row. Let $U \in \{0, 1\}^{m_2 \times d}$ be the universal signature matrix (see [4, 38] for definition). If $t_i$ is the $i$the row of $T_G = [t_1^T, \ldots, t_{m_1}^T]^T$, then the GT matrix A is formed as $A = [A_1^T, \ldots, A_{m_1}^T]^T$, where the submatrix $A_i = U diag(t_i)$ of size $m_2 \times d$. The total tests will be $m = m_1 \cdot m_2$. We have the following recovery guarantee of this construction:

**Proposition 1.** *Suppose we wish to recover a $k$ sparse binary vector $y \in \{0, 1\}^d$. A binary testing matrix $A$ formed from the regular-SAFFRON graph with $m = \tau_1.k \log \frac{d}{k}$ tests recovers $1 - \varepsilon$ proportion of the support of $y$ correctly with high probability (w.h.p), for any $\varepsilon > 0$. With $m = \tau_2 k \log k \log \frac{d}{k}$, we can recover the whole support set w.h.p. The constants $\tau_1$ and $\tau_2$ depend on $c, r$ and the error tolerance $\varepsilon$. The computational complexity of the decoding scheme will be $O(k \log \frac{d}{k})$.*

Proof of the proposition can be found in [38]. The decoding algorithm was discussed in the main text.

## B  Proof of Theorem 1

Next, we sketch the proof of Theorem 1 in the main text.

*Proof.* Let us denote the entries of $\tilde{H}$ and $A$ as $\tilde{h}_{i,j}$ and $a_{i,j}$ respectively, $i = 1, \ldots, m; j = 1, \ldots, n$. From our construction: $\Pr(a_{i,j} = 1) = \tilde{h}_{i,j}$ and $\Pr(a_{i,j} = 0) = 1 - \tilde{h}_{i,j}$.

First, let us find the probability that $z_i = 0$. Since $z_i$ will be 0 if and only if the support of $i$th row of $A$ has no intersection with the support of $y$, hence,

$$\Pr(z_i = 0) = \prod_{j \in \text{supp}(y)} \Pr(a_{i,j} = 0) = \prod_{j \in \text{supp}(y)} (1 - \tilde{h}_{i,j}).$$

Now note that, $b_j = \sum_{i=1}^{m} a_{i,j} z_i$. Therefore, $\mathbb{E}[b_j] = \sum_{i=1}^{m} \mathbb{E}[a_{i,j} z_i] = \sum_{i=1}^{m} \Pr(a_{i,j} z_i = 1)$. It turns out that,

$$
\begin{aligned}
\Pr(a_{i,j} z_i = 1) &= \Pr(a_{i,j} = 1, z_i = 1) \\
&= \Pr(a_{i,j} = 1) \Pr(z_i = 1 \mid a_{i,j} = 1) \\
&= \tilde{h}_{i,j}(1 - \Pr(z_i = 0 \mid a_{i,j} = 1)).
\end{aligned}
$$

Now, we consider two cases. When $j \in \mathrm{supp}(y)$, $\Pr(z_i = 0 \mid a_{i,j} = 1) = \Pr(\forall j \in \mathrm{supp}(y), a_{i,j} = 0 \mid a_{i,j} = 1) = 0$. On the other hand, when $j \notin \mathrm{supp}(y)$, $\Pr(z_i = 0 \mid a_{i,j} = 1) = \Pr(z_i = 0) = \prod_{l \in \mathrm{supp}(y)}(1 - \tilde{h}_{i,l})$. Therefore,

$$
\Pr(a_{i,j} z_i = 1) = \begin{cases} \tilde{h}_{i,j}, & j \in \mathrm{supp}(y) \\ \prod_{l \in \mathrm{supp}(y)}(1 - \tilde{h}_{i,l}) & j \notin \mathrm{supp}(y). \end{cases}
$$

Hence, when $j \in \mathrm{supp}(y)$,

$$
\mathbb{E}[b_j] = \sum_{i=1}^{m} \Pr(a_{i,j} z_i = 1) = \sum_{i=1}^{m} \tilde{h}_{i,j} = c.
$$

But when $j \notin \mathrm{supp}(y)$,

$$
\begin{aligned}
\mathbb{E}[b_j] &= \sum_{i=1}^{m} \Pr(a_{i,j} z_i = 1) \\
&= \sum_{i=1}^{m} \prod_{l \in \mathrm{supp}(y)} (1 - \tilde{h}_{i,l}) \leq \sum_{i=1}^{m} \prod_{l \in \mathrm{supp}(y)} \exp(-\tilde{h}_{i,l}) \\
&= \sum_{i=1}^{m} \exp\Big(- \sum_{l \in \mathrm{supp}(y)} \tilde{h}_{i,l}\Big) = \sum_{i=1}^{m} \exp(-\langle y, \tilde{h}^{(i)}\rangle). \qquad \square
\end{aligned}
$$

We can make stronger claims to bolster this theorem. Since the random variables $b_j, j = 1, \ldots, n$ are all Lipschitz functions of independent underlying variables, by using McDiarmid inequality [25] we can say that they are tightly concentrated around their respective average values.

## C   Additional experimental results

Here, we present additional results and further discuss the results we presented in the main text for the proposed methods. We then give few results which help us better understand the parameters that affect the performance of our MLGT method. First, we describe the evaluation metrics used in the main text and here for comparison.

**Evaluation metrics:** To compare the performance of the different MLC methods, we use the most popular evaluation metric called $Precison@k$ (P@k) [1] with $k = \{1, 3, 5\}$. It has been argued that this metric is more suitable for modern applications such as tagging or recommendation, where one is interested in only predicting a subset of (top $k$) labels correctly. P@k is defined as:

$$
P@k := \frac{1}{k} \sum_{l \in rank_k(\hat{y})} y_l,
$$

where $\hat{y}$ is the predicted vector and $y$ is the actual label vector. This metric assumes that the vector $\hat{y}$ is real valued and its coordinates can be ranked so that the summation above can be taken over the highest ranked $k$ entries of $\hat{y}$. For the hierarchical approach, we weight and rank the labels based on repeated occurrence (in the overlapping set $S$).

In general, MLGT method returns a binary label vector $\hat{y}$ of predefined sparsity, there is no ranking among its non-zero entries. Hence, we also use a slightly modified definition:

$$
\Pi@k := \frac{1}{k} \min\Big(k, \sum_{l \in top_5(\hat{y})} y_l\Big), \tag{3}
$$

where $top_5(\hat{y})$ is the 5 nonzero co-ordinates of $\hat{y}$ predicted by MLGT assuming that the predefined sparsity is set to 5. To make the comparison fair for other (ranking based) methods, we sum over the top 5 labels based on their ranking (i.e. we use $rank_5$ instead of $rank_k$ in the original definition).

Figure 4: $\Pi@1$ and $\Pi@3$ for test data instances for bibtex (top two) and RCV1x (bottom two) datasets as a function of number of groups $m$. Error bar over 10 trials.

Table 4: Comparisons between GT constructions. Metric: Modified Precision

| Dataset | Metrics | NMF - GT | CW - GT | SP - GT | OvA |
|---------|---------|----------|---------|---------|-----|
| Bibtex | $\Pi@1$ | **0.7354** | 0.7089 | 0.6939 | 0.6111 |
| $d = 159$ | $\Pi@3$ | **0.3664** | 0.3328 | 0.3034 | 0.2842 |
| $m = 120$ | $\Pi@5$ | **0.2231** | 0.2017 | 0.1823 | 0.1739 |
| | $\Phi_Y(A)$ | **10.610** | 12.390 | 12.983 | — |
| | $T_{total}$ | 5.13s | 4.01s | **3.98s** | 8.22s |
| | $T_{test}$ | **0.13s** | **0.13s** | **0.13s** | 0.18s |
| Mediamill | $\Pi@1$ | **0.8804** | 0.8286 | 0.6358 | 0.8539 |
| $d = 101$ | $\Pi@3$ | **0.6069** | 0.5413 | 0.2729 | 0.5315 |
| $m = 50$ | $\Pi@5$ | **0.3693** | 0.3276 | 0.1638 | 0.3231 |
| | $\Phi_Y(A)$ | **10.377** | 11.003 | 10.876 | — |
| | $T_{total}$ | 17.2s | **15.7s** | 15.82s | 29.4s |
| | $T_{test}$ | **0.17s** | **0.17s** | **0.17s** | 0.54s |
| RCV1x | $\Pi@1$ | **0.9350** | 0.9205 | 0.8498 | 0.9289 |
| $d = 2016$ | $\Pi@3$ | **0.6983** | 0.6596 | 0.5732 | 0.6682 |
| $m = 250$ | $\Pi@5$ | 0.4502 | 0.4104 | 0.3449 | **0.4708** |
| | $\Phi_Y(A)$ | **53.916** | 58.459 | 58.671 | — |
| | $T_{total}$ | 88.4s | 77.5s | **74.2s** | 363.2s |
| | $T_{test}$ | 1.20s | **1.04s** | 1.10s | 6.37s |
| Eurlex | $\Pi@1$ | 0.8477 | 0.8430 | 0.6792 | **0.8535** |
| $d = 3993$ | $\Pi@3$ | 0.5547 | 0.5582 | 0.3933 | **0.6132** |
| $m = 350$ | $\Pi@5$ | 0.3444 | 0.3597 | 0.2758 | **0.4085** |
| | $\Phi_Y(A)$ | 80.023 | 80.732 | 82.257 | — |
| | $T_{total}$ | 227.3s | 99.6s | **90.4s** | 560.1s |
| | $T_{test}$ | **0.94s** | **0.93s** | **0.93s** | 7.26s |

**Comparing group testing constructions:** In Table 4, we compare the three constructions discussed in this paper on four datasets. We also include the One versus All (OvA) method (which is computationally very expensive) to provide a frame of reference. In the table, we list $P@k$ for $k = \{1, 3, 5\}$, the correlation metric $\Phi_Y(A)$, the total time $T_{total}$ as well as the time $T_{test}$ taken to

Figure 5: Analysis: (Left) Relation between P@k and the correlation metric $\Phi_Y(A)$, (Middle) Relation between P@k and column sparsity c, and (Right) Performance of NMF for larger $m$.

predict the labels of $nt$ test instances. Figure 4 is the enlarged version of Figure 1 in the main text, where a smaller version of the plots were given due to space constraint.

The NMF-GT method performs better than both methods, because it groups the labels based on the correlation between them. This observation is supported by the fact that the correlation metric $\Phi_Y(A)$ of NMF-GT is the lowest among the three methods. Also note that even though NMF-GT has longer training time compared to the other GT methods (due to the NMF computation), its prediction time is essentially the same. We also note that the runtimes of all three MLGT methods are much lower than OvA, particularly for larger datasets as they require much fewer ($O(\log d)$) classifiers.

**Results discussion:** In table 2 of main text, we summarized the results obtained for six methods for different datasets. We note that NMF-GT performs very well given its low computational burden. PfastreXML and Parabel, on the other hand, yield slightly more accurate results but require significantly longer run times.

Note that, when compared to the MLGT, the other methods require significantly more time for training. This is because, the tree based methods use k-means clustering recursively to build the label tree/s, and require several OvA classifiers to be trained, one per each label in the leaf nodes. OvA methods are obviously expensive since they learn $d$ number of classifiers. Moreover, the prediction time for MLGT is also orders of magnitude less than many of the popular methods. In addition, the other methods have several parameters that need to be tuned (we used the default settings provided by the authors). We also note that the main routines of most other methods are written in C/C++ language, while MLGT was implemented in Matlab and hence the run times can be further improved to enable truly real-time predictions.

In Table 3 of the main paper, for the large two datasets, the label set was divided into blocks of sizes roughly around $40K$. We also used negative sampling of the training data for each block as done in many recent XML works [28, 15]. We also reduced the feature dimension via. sketching. For hierarchical partitioning, we used the vertex separator approach described in the main text, using the FORTRAN code provided by the author of [11]. The reordering for the four datasets in Table 3 are given in Figure 1 for the main text. The approach is extremely fast, and the runtime for the four datasets for reordering and partitioning were:

*Eurlex: 0.5s; Wiki10: 4.11s; WikiLSHTC: 40.3s; and Amazon670: 15.5s.*

For Eurlex and Wiki10, the accuracy and runtime results for SLEEC, PfastreXML and Parabel were computed by us using their matlab codes. Results for these three methods for the remain two datasets, and all results for the additional four methods (Dismec, PPD-sparse, XT and XML-CNN) were obtained from [28] and [40]. All runtimes are based on single core implementation.

**MLGT Analysis:** We conducted several numerical tests to analysis the performance of MLGT with respect to various settings. Figure 5 presents few of these numerical analysis results, which help us understand the performance of MLGT better. In the left figure, we plot the P@k achieve by MLGT with different GT constructions, as a function of the the correlation metric $\Phi_Y(A)$. The different points (circle) in the plot correspond to different GT matrices with different $\Phi_Y(A)$. These GT matrices were formed by randomly permuting $k$-disjunct matrices, and changing its size. We observe that GT matrices with lower $\Phi_Y(A)$, yield better classification. These results motivated us to develop the data-dependent grouping approach.

Table 5: Average Hamming loss errors in reduction v/s training

| Dataset | $d$ | NMF-GT | | CW-GT | |
|---|---|---|---|---|---|
| | | R-Loss | T-Loss | R-Loss | T-Loss |
| Bibtex | 159 | 3.49 | 3.68 | 2.95 | 4.30 |
| RCV1-2K | 2016 | 3.99 | 4.72 | 3.96 | 4.91 |
| EurLex-4K | 3993 | 1.38 | 4.77 | 1.05 | 5.03 |

Figure 6: Prec@1 for test data instances for bibtex (left) and RCV1x (right) datasets as a function of fraction of training data used. Error bar over 5 trials

In the middle plot, we have the performance of the NMF-GT method for different column sparsity $c$. We clearly note that as $c$ increases, the performance first increases, and then reduces for larger $c$. This is because, for larger $c$, the GT matrix will have higher coherence between the columns. As indicated in our analysis, the performance of the GT construction will depends on this coherence. This analysis motivated us to use the search technique described in Remark 1, to select the optimal column sparsity $c$.

In the right plot, we compare the performance of NMF-GT vs CW-GT as a function of number of groups $m$ for the Eurlex dataset. We observe that for smaller $m$, NMF-GT performs better. However, for larger $m$ and more so for larger number of label $d$, NMF-GT becomes less accurate. This is due to the difficulty in computing accurate NMF for such large matrices. NMF is known to be an NP hard problem. This result likely explains why the NMF-GT's performance on larger datasets is less accurate. A possible approach to improve the accuracy of NMF-GT is to use the Hierarchical approach described above and split the large label set into smaller disjoint subsets, and apply NMF-GT independently.

In table 5, we list the average Hamming loss errors we suffer in label reduction (and decoding) when using NMF-GT and CW-GT for the three datasets. That is, we check the average error in the group testing procedure (label reduction and decoding), without classifiers. We also list the average Hamming loss in the training data after classification for comparison. We observe that, the NMF-GT has worse reduction loss compared to CW-GT. This is because, NMF-GT is data dependent, and is not close to being k-disjunct as oppose to CW-GT, which is random. However, we note that the training loss of NMF-GT is better. This shows that, even though the reduction-decoding is imperfect (introduces more noise), NMF-GT results in better individual classifiers. These comparisons show that data-dependent grouping will indeed result in improved classifiers.

**Implementation details:** All experiments for NMFGT and He-NMFGT were implemented in Matlab, and conducted on a standard work station with Intel i5 core 2.3GHz machine. The timings reported were computed using the `cputime` function in Matlab. For the SLEEC method, we could not compute $\Pi@k$ as in eq. 3, since the source code did not output the score matrix. The $\Pi@k$ reported for SLEEC in Table 4 were the P@k returned by source code. Also, for the last 2 examples, SLEEC was run for 50 iterations (for the rest it was 200).

# D  Learning with less training data

In supervised learning problems such as MLC, training highly accurate models requires large volumes of labeled data, and creating such volumes of labeled data can be very expensive in many applications [21, 41]. As a result, there is an increasing interest among research agencies in developing learning algorithms that achieve 'Learning with Less Labels' (LwLL)[2]. Since MLGT requires training only $O(k \log d)$ classifiers (as opposed to $d$ classifiers in OvA or other methods), we will need less labeled data for training the model. In section 5, we present preliminary results that demonstrate how MLGT achieves *learning with less data* for MLC.

Here, we present preliminary results that demonstrate how MLGT achieves more accurate (higher precision) with less training data compared to the OvA method (see Table 4 in suppl). Figure 6 plots the precision (Prec@1) for test data instances for the bibtex (left) and RCV1x (right) datasets, when different fractions of training data were used to train the MLGT and OvA models. We note that MLGT achieves the same accuracy as OvA with only 15-20% of the number of training points (over $5\times$ less training data). We used the same binary classifiers for both methods, and MLGT requires only $O(k \log d)$ classifiers, as opposed to OvA, which needs $d$ classifiers. Therefore, MLGT likely requires fewer training data instances.

## Footnotes

[2]darpa.mil/program/learning-with-less-labels