[Reviews · NeurIPS 2020]

Review 1

Summary and Contributions: This paper applies a previously published Multi-Label Group Testing (MLGT) given in [35] and make some improvements over the existing one. To this end, the authors propose two methods to construct the group testing matrix, A, by using symmetric nonnegative matrix factorization (symNMF) and hierarchical partition. The method using symNMF yields very poor results compared to state-of-the-art methods whereas the one using hierarchical approach improves the state-of-the-art results on a single dataset.

Strengths: The strengths of the paper can be summarized as follows: 1) The authors extend the MLGT method described in [35] in two ways for constructing the group testing matrix A: In the first approach, the authors apply symmetric nonnegative matrix factorization to the matrix, YY^t, where Y indicates the label matrix. In the second approach, the authors use a 2) Some improvements over state-of-the-art are obtained in one dataset.

Weaknesses: The weaknesses of the paper can be summarized as follows: 1) First of all, the authors build their method based on MLGT given in [35]. In fact, the algorithms given in the paper are very similar to the ones given in [35] (training algorithm is identical whereas the prediction algorithm slightly differs). Therefore, the novelty of the method is limited. 2) The authors pus too much effort on describing the construction of the group testing matrix A by using label matrix, yet it is not practical since one has to work on very large matrices as admitted by the authors. Furthermore, the accuracies of this procedure is very low compared to the state-of-the-art methods. Hierarchical approach yields much better accuracies, so I wonder why the authors put too much effort with defining it. Instead, hierarchical approach needs more explanation. 3) Experimental study is weak and confusing. First of all, the authors emphasize the results of NMF-GT and states that it outperforms other two competing methods. However, one of them is the baseline given in [35] and the other is also their proposed alternative, and these accuracies are very low compared to the state-of-the-art as seen in Table 2. Furthermore, the results given SP-GT do not match the results obtained in [35]. 4) Hierarchical approach yields much better accuracies compared to NMF-GT. I wonder why the author did not test in on the first two datasets given in Table 2. The same approach only beats some state-of-the-art results on modified precision for k=1 on Table 2 and it is significantly outperformed by the competing methods for k=3 and k=5 on Eurlex and Amazon13 datasets. It slightly beats the other methods on Wiki10 dataset only in Table 2. But, in Table 3, it is clear that the proposed method is significantly outperformed by the other-state-of-the art methods. Thus, the contribution of the method is questionable.

Correctness: The authors build their method on a published methodology and most of the claims come from there. Therefore, they must be correct. The authors only make some changes to create a different group testing matrix which beats the one introduced in the original paper.

Clarity: The paper is not written well since the reader is referred to the supplementary material frequently for details.

Relation to Prior Work: The authors build their method based on MLGT method introduced in [35]. Here, teh authors introduce new techniques for creating the group testing matrix A. The differences are clear.

Reproducibility: No

Additional Feedback: This paper may have some potential but it is not ready for the publication yet since the experimental study is very weak. The accuracies of the proposed methods are very low compared to the state-of-the-art methods.


Review 2

Summary and Contributions: The paper proposes an extension of the group testing methods (GT, [35]) for multilabel classification. GT framework consists in defining a set of labels subsets - the group testing matrix - and a set of binary classifiers to identify the subset. The main contributions of the paper are a new construction algorithm for the group testing matrix, taking into account label co-occurences and the use of a fast decoding process adapted from the state of the art. Authors propose to build the testing matrix using a symmetric non negative matrix factorization and to sample over the factorized matrix to ensure an average of k labels by subset. As NMF could be costly for extremely large label sets, the authors propose to partition the data using an hierarchical based partitioning (using the assumption that in extreme classification dataset few labels are interacting with many others, most labels in co-occurrence are in small clusters). The NMF is applied to each cluster of co-occuring labels. The second contribution to ensure a fast decoding is a straight adaptation of an existing algorithm. The approach is assess on real datasets and compared to SOA multilabel classification algorithms. *** After feedback : Thanks to the authors for the clarification and the additional experiments.

Strengths: The paper proposes an improvement over the group testing framework to build the group matrix and for the decoding step. The experiments asses the relevance of the approach for large scale multi-label dataset. The experiments also show that taking into account the label correlation is a key factor for extreme classification task. The results do not improve the SOA in terms of precision but the time complexity of the proposed method is impressive, approaching logarithmic inference time of the tree based approaches.

Weaknesses: The paper do not introduce a real novel approach, the presented work uses existing approaches and combines them in order to achieve better performances. It solves draw back of previous approaches using different tools from the recent literature.

Correctness: The methodology is correct and the proofs seem sound.

Clarity: The paper is dense and some parts should be hard to follow for non specialist (especially section 2 and section 4), but overall the paper is self-contained and the reading of the annex is not required to understand the paper.

Relation to Prior Work: SOA is clearly discussed and improvements are clearly stated. The only missing main SOA algorithm in table 3 is AnnexML (Tagami 17), it would be interesting to include the results as it has very good performances for a competitive time inference.

Reproducibility: Yes

Additional Feedback: Two remarks : * it is usual in extreme classification context to use ensemble methods to improve the result of a single classifier, did you try ? * some details are missing to reproduce the results : what kind of binary classifiers you used ? how to choose hyperparameters (number of subsets for instance in hierarchical partitioning ) ?


Review 3

Summary and Contributions: The paper presents an embedding based method for multi-label classification which is scalable to large number of labels by grouping the labels via a data-dependent scheme. It builds on a recent idea of using group testing for multi-label classification. While being much faster to train and predict the unknown test labels, the proposed method achieves somewhat comparable results on benchmark datasets. *******After author feedback ************* Thanks to the authors for the comments. I update my rating accordingly.

Strengths: - The proposed method is well-motivated, and seems theoretically sound with connections to approaches in group testing. In this regard, the proposed method begins in new perspective for the domain of large-scale multi-label classification - Though the empirical results in terms of prediction performance are not very strong, the method has dvantages in terms of training and prediction effiiciency compared to most state of the art methods.

Weaknesses: - The performance of the method is still sub-optimal on large scale datasets, and hence there is a scope of improvement. - The proposed method is somewhat similar to the earlier work [35] in terms of overall training and prediction algorithms, and some finer details such as usage of data-dependent methodologies for grouping is used.

Correctness: The proposed method seems correct in terms of the main idea, and the code/steps to recreate the results are also provided.

Clarity: Though the paper is clear in most parts, the figures are not clearly readable such as Figure 2 in the paper, and others in the appendix

Relation to Prior Work: The paper does a fine job in position the work wrt to earlier works.

Reproducibility: Yes

Additional Feedback: As that mentioned above under 'Clarity' and 'Weaknesses' heading.


Review 4

Summary and Contributions: This work improves upon a previous work that proposed to use ideas from group testing to extreme classification. The proposal leverages recent results in group testing and decoding for obtaining label groups in a data-dependent fashion and leading to log-time fast inference. Further, label correlation is used to decompose the problem so that the idea can be employed in each simultaneously.

Strengths: 1. Though the work is incremental it seems to leverage various recent ideas in group testing to improve on the predecessor. This is nice and definitely brings in fresh thoughts to extreme classification literature. 2. The algorithm seems to scale really well to large extreme classification benchmarks.

Weaknesses: 1. According to me the simulations results clearly show that there exists a trade-off. It seems the proposed methodology achieves speed-up, many a time, by loosing ground in terms of generalization. Hence it would be very interesting to know if some knobs/hyper-parameters can be varied to achieve various trade-off points. For e.g., a time vs accuracy kind of plot for various hyper-parameters would be more illustrative of the merit of the proposal. 2. In section4, when solutions from the various partitions are merged, normalization issues may occur. For e.g., the scores may not be comparable across partitions. Discussion regarding this seems to be completely missing. Minor: 3. The idea in section 4 is ok, but I think it is all about using the label correlation information rather than any hierarchical information. So simply naming it so would be ideal and infact more generic as correlations may occur even when there is no hierarchy. Also, is there an automatic way to figure out when there may not exist well-defined partitions in the labels (as per correlations)? Currently, this seems to be done via manual inspection.

Correctness: The methods seem to be correct and some issues in simulations are discussed above.

Clarity: Readability is ok. Replacing "hierarchical" with "label-correlations" in sec4 seems more appropriate as discussed above.

Relation to Prior Work: yes.

Reproducibility: Yes

Additional Feedback: I have read the feedback and my queries were answered satisfactorily. So I am increasing the score.

[Author Response · NeurIPS 2020]

We thank the reviewers for their valuable time and thoughtful feedback. We are encouraged by the positive comments
w.r.t. our novel ideas and application of group testing to multilabel classification (MLC)([R3],[R4]), our method's
scalability ([R4]), impressive runtimes ([R2]), and efficiency ([R3]). Please see our response below:

[R1],[R2],[R3] **Novel Contributions:** Our main contributions are: (a) the development of a (non-trivial) data-dependent
group testing scheme, that improves label grouping for MLGT significantly (vs. [35]), and can use the recently proposed
log-time decoding algorithm; (b) the use of matrix reordering techniques to hierarchically partition the label space, so
that we can apply MLGT to subsets of labels independently, in order to scale to very large label sets. These innovations
lead to a significantly faster training algorithm (Table 3) compared to most existing methods ($\sim 50$min vs. 370-730 hrs
for DISMEC, which has the highest accuracy), yet yield comparable results. Note that (more accurate) OvA methods
require $O(d)$ classifiers to be trained (taking many hours). The tree methods use k-means clustering to create label
clusters (we use fast matrix reordering heuristics) and OvA classifiers at each leaf nodes (# classifiers $\sim o(d)$ vs. our
$O(k \log d)$). Our method also has a provably log-time prediction algorithm, enabling almost real-time predictions.

[R1],[R2],[R3] **Similarity to [35]:** As we build on the MLGT algorithm of [35], where group testing was first proposed
for MLC, the core idea of creating $O(k \log d)$ label groups is similar. However, the method in [35] yields poor accuracy
on large datasets due to random label grouping, and does not scale to extreme settings. We overcome these issues by first
developing a data-dependent grouping scheme (NMFGT) to improve the method's accuracy, and then use hierarchical
partitioning to scale the method to very large problems. Sampling a group testing matrix that (a) captures the label
correlations, (b) has distinctive columns, and (c) satisfies the SAFFRON construction, is non-trivial. We propose a
technique that uses a normalized NMF basis (capturing label correlations) as a potential function for sampling columns
of the GT matrix that are distinct and have fixed average sparsity $c$ (left regular graph). Computing a good GT matrix
via NMF for large $d$ is difficult, and we use label partitioning in order to apply NMF-based MLGT to smaller problems.

[R1] **Weakness 1,2**: We develop the NMFGT and *also* the HE-NMFGT (NMFGT + hierarchical partitioning) to (a)
tackle large datasets and (b) get better accuracy on large datasets, and thus describe both. **Weakness 3 - Experimental**
**study**: We first show that NMFGT is better (See Fig 2. & suppl.) than earlier GT method in [35]. Also, SP-GT
results do not match with [35] as the modified prediction algorithm in this paper is better (we get better accuracy than
[35,Table 1]. We next use label partitioning to improve over NMF-GT for larger datasets (Table 2). Finally, we show
that partitioning+NMFGT has *significantly faster* training and prediction times than other methods (Table 3). We
believe that low training times (saving many hours) and fast predictions in return for a limited loss (few points) in
accuracy will be critical in many "related search" applications. **Weakness 4**: We use HE-NMFGT only when # labels is
too large to apply NMFGT. We do mention that for Mediamill and RCV1x there were no clear label partitions.

[R3], [R4] **Trade off and improvements:** We thank the reviewers for these suggestions. In the figures below we plot
precisions ($\Pi@1, \Pi@3, \Pi@5$) versus runtimes (in secs) for Eurlex (left) and wiki10 (middle) datasets, by increasing
# groups $m$ in each partition. Indeed, we notice a clear trade-off: as we increase runtimes, accuracy improves. But
beyond a point, the accuracy gain is limited as $m$ is increased. In the paper, we chose smallest $m$ (vertical line) for
which our accuracy is close to the SOA tree methods. Improved accuracy can be achieved for higher runtimes (when
$m$ is much more than $k \log d$). We also plot $\Pi@k$ versus # partitions $\ell$ for Wiki10 (right). For smaller $\ell$, it is hard to
compute a good NMF for large matrices, and with many partitions, we will miss certain label correlations.

[R4] **Solution merger:** The output of MLGT will be a binary vector $\{0, 1\}^{d_i}$, hence, comparing scores across disjoint
subsets of labels will not be an issue. For the shared labels across partitions, we indeed use weights for the label outputs
such that these weights add to 1. Due to space constraints these details were only briefly discussed (in sec. 4). **Label**
**partitions:** The matrix reordering method recursively partitions the labels, hence discovering a hierarchy. The code we
use produces the partitions (and sizes), in addition to the permutations depicted in Fig. 1. So the process is automatic.

[R2] **Ensemble methods, missing details/comparisons:** The ensemble idea is an exciting direction we have not
investigated! Linear SVM was used for classifiers (same as SOA tree methods). For large datasets the partition sizes
were $\sim 40k$ labels. We had to defer the implementation details to supplement due to space constraints. We will include
a comparison to AnnexML (Tagami, 17) if the paper is accepted.

[Meta-Review · NeurIPS 2020]

The authors introduce a method for speeding up a group testing approach for multi-label classification. Thanks to this, the new algorithm can be used for problems with a large number of labels. The paper is clearly written. Some reviewers were underlining the incremental contribution, but majority of them agreed with the authors that improving complexity of the existing solution is a sufficient contribution. The authors should, however, revise their discussion on the complexity of clustering methods used by the label tree approaches. Certainly the variants of k-means used there are not scaling quadratically with the number of labels!